# Polyamide Microparticles with Immobilized Enological Pectinase as Efficient Biocatalysts for Wine Clarification: The Role of the Polymer Support

**DOI:** 10.3390/molecules30010114

**Published:** 2024-12-30

**Authors:** Sandra C. Oliveira, Samuel M. Araújo, Nadya V. Dencheva, Zlatan Z. Denchev

**Affiliations:** IPC-Institute for Polymers and Composites, University of Minho, 4800-056 Guimarães, Portugal; b8223@dep.uminho.pt (S.C.O.); a85216@alunos.uminho.pt (S.M.A.)

**Keywords:** wine clarification, pectinase, enzyme immobilization, anionic polymerization of lactams, biocatalyst reusability, PA4, PA6, PA12

## Abstract

Free pectinase is commonly employed as a biocatalyst in wine clarification; however, its removal, recovery, and reuse are not feasible. To address these limitations, this study focuses on the immobilization of a commercial pectinolytic preparation (Pec) onto highly porous polymer microparticles (MPs). Seven microparticulate polyamide (PA) supports, namely PA4, PA6, PA12 (with and without magnetic properties), and the copolymeric PA612 MP, were synthesized through activated anionic ring-opening polymerization of various lactams. Pectinase was non-covalently immobilized on these supports by adsorption, forming Pec@PA conjugates. Comparative activity and kinetic studies revealed that the Pec@PA12 conjugate exhibited more than twice the catalytic efficiency of the free enzyme, followed by Pec@PA6-Fe and Pec@PA4-Fe. All Pec@PA complexes were tested in the clarification of industrial rosé must, demonstrating similar or better performance compared to the free enzyme. Some immobilized biocatalysts supported up to seven consecutive reuse cycles, maintaining up to 50% of their initial activity and achieving complete clarification within 3–30 h across three consecutive cycles of application. These findings highlight the potential for industrial applications of noncovalently immobilized pectinase on various polyamide microparticles, with possibilities for customization of the conjugates’ properties.

## 1. Introduction

Wine is one of the oldest and most widely consumed alcoholic beverages, produced through the enzymatic alcoholic fermentation of grape juice or must. Most of the enzymes required for this process are derived from the grapes themselves, the grape’s natural microflora, and/or microorganisms involved in fermentation [1,2]. However, in modern winemaking, particularly at large scales, the endogenous enzymes from grapes, yeasts, and naturally occurring microorganisms often prove insufficient to catalyze the numerous chemical reactions involved in the fermentation process. Consequently, the use of commercial enzyme supplements at various stages of winemaking has become common practice in large-scale production [3,4].

Clarification is a crucial step in vinification, aimed at eliminating turbidity from the final product. This turbidity is primarily caused by high-molecular-weight pectic substances originating from grape cell walls, which are transferred into the must. In rosé wine production, the addition of exogenous pectinolytic enzymes is essential not only to enhance the aesthetic and sensory qualities of the wine but also to improve vinification efficiency [5].

The most commonly used enzymes for clarification are pectinases, a heterogeneous group of enzymes that catalyze the controlled degradation of pectin, the key substance responsible for turbidity. Beyond clarifying the must, the enzymatic breakdown of pectin offers several additional benefits: (i) it reduces must viscosity, facilitating downstream processes, e.g., minimizing filtration system clogging [6,7]; (ii) it increases juice yield by up to 15% [8]; and (iii) it can release volatile aroma compounds, particularly terpenes and esters trapped within the pectin structure, thereby enhancing the wine’s aromatic profile and sensory complexity [9].

Pectinases used in the winemaking industry are typically applied as preparations, i.e., enzyme mixtures derived from fungal sources, particularly species of *Aspergillus* and *Penicillium*. The primary enzymes in these preparations are polygalacturonase (E.C. 3.2.1.15), pectin lyase (E.C. 4.2.2.10), and pectin methyl esterase (E.C. 3.1.1.11), which work synergistically during the clarification step to degrade pectin into simpler carbohydrates [10].

Currently, industrial enological pectinases are mainly used as free enzymes, i.e., they are soluble in the aqueous reaction medium. However, in this form, they often exhibit insufficient short-term operational stability and are difficult to recover and reuse [11], complicating the automation and continuous operation of enzymatic processes [12]. To overcome these limitations, the immobilization of enzymes has been widely studied, as highlighted by recent reviews disclosing various immobilization strategies [2,13,14]. In summary, immobilization involves attaching enzymes to various carriers, frequently selected from the group of natural or synthetic polymers. This can be achieved through physical methods (adsorption and entrapment) or chemical reactions (covalent binding and crosslinking). These approaches improve enzyme stability, facilitate the removal of biocatalysts from the final product, enable enzyme reuse, and enhance the development of continuous industrial processes.

In the food industry in general, and particularly in winemaking, supports for enzyme immobilization must be nontoxic and biocompatible, ensuring that they do not alter the sensory or chemical properties of the final product, such as color, taste, or texture [15]. The enzyme/support complex must be robust enough to withstand external factors encountered during processing, such as filtration, mixing, and exposure to temperatures or pH levels that are unusual for the free enzyme. A large surface area is crucial for maximizing enzyme loading and promoting efficient mass transfer during catalysis. Finally, the supports should enable cost-effective catalytic complexes with high activity while being easy to recover and reuse across multiple operation cycles [16].

For the particular case of pectinase immobilization, a number of polymer-based supports have been reported, e.g., alginate [11], polyacrylamide [17], and polyethyleneimine gels [18], as well as cationic polystyrene resin beds [19], among others. Special attention has been given to polyamides (PAs), the closest synthetic analogues of the protein apoenzyme matrix. Polyamides are readily obtainable in various forms via ring-opening polymerization or polycondensation. Thus, Omelková et al. [20] and Rexová-Benková et al. [21] used PA6 powders as supports for polygalacturonase covalently immobilized via glutaraldehyde binding. The activity of the immobilized enzyme decreased by half, most likely due to steric hindrance near its active site [21]. Shukla et al. [22] immobilized polygalacturonase on PA6 beads using a similar method, achieving 70 μg/g enzyme loading, while retaining almost 50% activity after four clarification cycles. Ben-Othman and Rinken [23] immobilized pectinolytic enzymes on PA66 pellets or threads, proving a 40-fold higher immobilization yield in the latter case. The immobilized pectinase retained 40% activity after five cycles and over 20% after twenty cycles. Other authors [24,25] also employed PA6 and PA66 pellets as supports for pectinases, with covalent binding via glutaraldehyde or dimethyl sulfate. Notably, these two reagents are far too toxic and are not approved in food processing.

In summary, synthetic polyamides could serve as useful supports for pectinase immobilization, provided that the reduced catalytic activity of the resulting conjugates is addressed, while also eliminating the use of toxic compounds for covalent binding. These challenges can be overcome by employing noncovalent immobilization of pectinases onto polyamide microparticles (MPs), enabling the formation of multiple hydrogen bonds between the enzyme and the highly porous, scaffold-like polyamide MPs. As demonstrated in our previous work [26], such porous PA6 microparticles, which may also possess magnetic susceptibility, can be synthesized through activated anionic ring-opening polymerization (AAROP) of ε-caprolactam. The effectiveness of PA6 as a carrier for the noncovalent immobilization of pectinase, particularly for enological applications, was recently confirmed [27]. Compared to the free enzyme, PA6-immobilized pectinase (Pec@PA6) exhibited more than double the specific activity toward pectin, showed slightly increased substrate affinity, and acted as faster catalysts with greater resistance to substrate inhibition. Additionally, the Pec@PA6 biocatalysts could be reused in up to three consecutive cycles, achieving complete rosé must clarification within 3–36 h, which is considered acceptable for industrial applications.

This study extends our previous approach that was limited only to PA6 by synthesizing other types of polyamide MPs (i.e., PA4, PA12, and the copolymeric PA612) with and without magnetic susceptibility using AAROP. These microparticles are used to noncovalently immobilize pectinase, resulting in Pec@PA catalytic conjugates. After structural and morphological characterization, their enzymatic activity and the kinetics of pectin breakdown in each case are evaluated for rosé must clarification. The results are compared with Pec@PA6 systems and free, non-immobilized pectinase. Based on these findings, the influence of the chemical structure and morphology of different polyamide MPs supports on the pectinolytic performance of the immobilized pectinase is determined. The best solutions for potential industrial applications are also identified.

## 2. Results

### 2.1. Synthesis of Different Polyamide MPs

Seven different MP supports were synthesized and designated as shown in Table 1 and explained in Section 3.3. The MPs without Fe were white powders, while those with Fe were gray and exhibited ferromagnetic properties. The main characteristics of these samples are also presented in the same table. The structures of the monomers and catalysts used in each synthesis, along with the resulting polymers, are shown schematically in Appendix A.

Table 1 shows that AAROP in suspension resulted in relatively good yields of purified polyamide MPs, with the highest yield for the copolymeric PA612 sample and the lowest for PA12 MP. In the case of PA12, the process was more difficult to control due to rapid spontaneous MP growth leading to the formation of large PA12 lumps instead of microparticles. To avoid this, the polymerization was stopped earlier, which explains the lower yield. In contrast, the high yield of the copolymeric system can be attributed to the stable growth of PA612 MP, allowing the process to be carried out at maximum temperature and stirring rates. Yields for the other MPs were around 60%. The viscosity-average molecular weights (*M_η_*) were around 30 kDa or slightly higher, sufficient for use as enzymatic supports. PA4 exhibited the lowest *M_η_*, likely due to the low polymerization temperature of 40 °C. Raising the temperature above 60 °C led to a marginal increase in molecular weight and very low yields of PA4, most probably related to spontaneous depolymerization due to unfavorable thermodynamics of AAROP in the case of five-membered polymerizable rings [28].

The Fe percentages of the PA-Fe supports were based on the amounts introduced into the reaction mixture before polymerization. As shown in previous thermogravimetric measurements [29], the actual Fe content in the final polyamide MPs (Table 1, values in brackets) depends on the degree of monomer conversion. At a conversion rate of 60%, the Fe content is typically twice that of the initial monomer feed.

### 2.2. Morphological Studies by SEM

The morphology of the polyamide MP support can be studied using SEM (Figure 1). For easier comparison, the micrographs of the neat support particles and the respective Pec@PA-immobilized complexes are presented side by side at the same magnification.

Several important conclusions can be drawn about the morphology of the neat supports and the corresponding pectinase-immobilized systems. The PA4 and PA4-Fe MPs exhibit predominantly near-spherical morphologies with diameters in the 10–40 µm range (Table 1, Figure 1a and Appendix A). PA4-Fe shows significantly larger pores, as seen in Appendix A). No iron particles are observed on the surface of the PA4-Fe MP, but they are found deeper inside the particles. As shown in Appendix A, the inset reveals that the two Fe peaks in EDX for K_α_ and K_β_ are clearly observable at 6.4 keV and 7.2 keV, directing the electron beam directly into a surface crack. After pectinase immobilization, the visual appearance of the Pec@PA4 complexes at lower magnification, both with and without Fe, remains largely unchanged (Figure 1a,c). At the micropore level, however, the enzyme fills in the pores, significantly altering the surface topology of Pec@PA4 (see Figure 1b,d).

The neat PA6 support consists of aggregates of irregular shape, up to 150 µm in size, formed by smaller fused spheres (Figure 1e,f). Introducing Fe results in the formation of a second MP fraction with higher aspect ratios, exhibiting pea-pod-like morphologies up to 300–500 µm in size (Appendix A), likely caused by the magnetic attraction of Fe microparticles followed by coating with the PA4 formed during AAROP. Before immobilization, the PA6 MP at higher magnification displays a scaffold-like structure composed of a 3D network of interlaced PA6 filaments (such as balls of yarn) with large meshes of 300–500 µm (Figure 1f). Enzyme immobilization does not significantly alter the form or size of the PA6 MP or the mesh filling (Figure 1g,h), with the enzyme appearing to adhere to the filaments without significantly obstructing the 3D network.

In contrast to PA4 and PA6, the neat PA12 MP lacks spherical symmetry (Figure 1i). At lower magnifications (Appendix A), most particles represent massive formations with irregular form, with sizes approaching 700 µm, and a finer fraction measuring 50–100 µm. The surface topography is sponge-like, featuring large, deep channels rather than pores (Table 1, Figure 1j). In PA12-Fe, these channels are even larger, but the size of the microparticles significantly decreases (Table 1, Appendix A). The presence of Fe in the AAROP reaction mixture most probably promotes heterogeneous nucleation of the forming PA12, facilitating its cold crystallization, which results in a better control over MP size. After immobilization, Pec@PA12 and Pec@PA12-Fe behave similarly, retaining the morphology of the neat support (Appendix A and Figure 1i–l), with smaller surface pores filled, while the large channels remain unobstructed. In the Fe-containing PA12 MP, small Fe particles may become visible on or slightly beneath the MP surface (Appendix A, the bright circular spot with clear EDX signals for Fe).

The copolymeric PA612 support exhibits a morphology more similar to PA12 than to PA6 MP, also with a bimodal size distribution. The predominant morphology consists of irregularly shaped particles up to 150 µm, with a sponge-like surface topography (Figure 1m,n). Pectinase immobilization covers all surface channels and pores (Figure 1o,p).

### 2.3. Noncovalent Immobilization of PEC on PA MP

The amount of noncovalently immobilized pectinolytic enzymes on both PA6 MP supports was calculated using Equation (3), Section 3.5. Figure 2a compares the UV/VIS spectra of the Viazyme stock solution used for immobilization (free Pec curve) with the remaining seven supernatants after the removal of the respective immobilized Pec@PA complexes. The absorption peak at λ = 263 nm corresponds to protein moieties containing substituted benzene rings, likely from tryptophan, phenylalanine, or tyrosine in the pectinase apoenzymes. As shown in previous studies [27], this direct spectroscopic method provides reliable data on protein adsorption immobilization, with a standard deviation of 3–5%, whereas the indirect protein determination by the Bradford and bicinchoninic acid assays can show data dispersion of up to 15%.

Figure 2b illustrates the immobilization yield of Pec@PA6 and Pec@PA6-Fe complexes, expressed as the percentage of pectinolytic enzymes immobilized relative to the initial concentration. Standard deviations were calculated from three consecutive trials for each sample. In Figure 2b, the word “Pec” is omitted for better readability.

Immobilization yield ranged from 13% to 18% for the PA4- and PA6-containing complexes and reached 38% for the PA12-Fe-supported pectinase, highlighting the significant influence of the polyamide support on the noncovalent immobilization of the enzyme. Two main factors most probably influence the amount of immobilized enzyme: (i) the density of hydrogen bonds between the proteic matrix of the pectinase apoenzyme and the polyamide support, and (ii) the surface topology of the respective MP carrier. Based on the data in Figure 2b, it appears that the second factor—the large pores and channels in the PA12 MP—plays a more important role than the H-bond density at the enzyme-support interface. This is evidenced by the fact that the PA4-based complexes, which should have the highest H-bond density, exhibit the lowest immobilization yield.

The formation of hydrogen bonds between the apoenzyme and the amide groups of the supports was investigated using FTIR. Figure 3 below and Appendix A provide representative spectra comparing selected polyamide MPs before and after pectinase immobilization.

The two spectra of PA6- and PA6-Fe supports in Figure 3a are identical and both exhibit bands at around 3300 cm^−1^, corresponding to the stretching vibrations in linearly associated secondary NH groups through H-bonds. Additionally, these spectra feature well-defined peaks for the Amide I band at 1631 cm^−1^ and Amide II at 1535 cm^−1^ with similar intensities, indicating a fixed trans-conformation of the NH–CO group, typical of high molecular weight polyamides. The free Pec spectrum (Figure 3a) proves the enzyme’s proteic matrix as a structural analogue of synthetic polyamides, with similar Amide I and Amide II peaks, also showing a broad peak centered at 3250 cm^−1^, representative of associated secondary and primary amino groups. An intensive band at 1050 cm^−1^, typical only for Pec, is also present. In the Pec@PA6 spectrum (Figure 3b), this band appears as a weak doublet at 1050–1120 cm^−1^, confirming the presence of Pec on the support. The increased intensity of the shoulder for associated –NH– and –NH_2_ groups suggests hydrogen bond formation between the enzyme’s amide groups and those of the PA6 support.

### 2.4. Activity by the DNS Methods

At temperatures near 100 °C, reducing sugars such as galacturonic acid (GA), which contain an aldehyde group, donate a hydrogen atom to the 3,5-DNS reagent (2-hydroxy-3,5-dinitrobenzoic acid), converting it into 3-amino-5-nitrosalicylic acid (3,5-ANS) while oxidizing GA to galactaric acid. This redox reaction causes a color change, as shown in Figure 4a, which serves as the basis for the UV-VIS estimation of pectinase activity [30].

In the present study, both free and immobilized pectinase are expected to depolymerize pectic substrates found in the rosé must, producing GA through the hydrolysis of the pectin main chain. The amount of GA generated will be proportional to the concentration of 3,5-ANS, which can be quantified via UV/VIS spectroscopy by the intensity of the band at λ = 540 nm. Thus, the pectinolytic activity is determined by the concentration of 3,5-ANS released per unit time using the calibration plot in Figure 4b.

It is important to emphasize that the activity assay should be made on the basis of the absorbance at 540 nm. As seen from Appendix A, the nominal 3,5-ANS peak becomes clearly resolved exactly at 540 nm only at higher concentrations of the reducing GA.

Figure 5 presents the results from the activity tests for free and immobilized pectinolytic preparations obtained at 23 ± 1 °C. All pectin-immobilized polyamide MP supports, both with and without Fe, are shown for comparison, the word “Pec” being omitted for better readability. As depicted in Figure 5a, almost all conjugates—except for Pec@PA4-Fe—demonstrate statistically equal absolute activity levels, which are approximately half that of the free enzyme. The small yellow bars in Figure 5a represent the measured “activity” of the polymer supports without any pectinase, showing values within the margin of the experimental error. This confirms that, as expected, the plain microparticulate supports do not possess pectinolytic activity.

After normalizing activity values by the protein content in each Pec@PA conjugate or in the free enzyme (Figure 5b), it becomes clear that the specific activities of the immobilized systems are either equivalent to (Pec@PA12-Fe) or 2.5–3.5 times higher than those of the free enzyme (Pec@PA4 and Pec@PA12, respectively). This result contrasts with the reduced activity often seen in covalently immobilized pectinase (e.g., in [21]), underscoring an important advantage of the noncovalent immobilization.

The observed significant increase in the recovered activity of adsorption-immobilized pectinolytic preparations on polyamide MPs may be due to a more favorable active site configuration and/or to reduced inactivation. To further investigate and validate the activity data, comparative kinetic studies were conducted on all biocatalysts in this study.

### 2.5. Kinetics Studies

The kinetic parameters of pectin hydrolysis catalyzed by free Pec and its seven noncovalently immobilized complexes were analyzed by plotting the reaction velocity per unit protein (*V_spec_*) against pectin substrate concentration (S). As demonstrated in Figure 6a, neither the free enzyme nor any of its immobilized complexes showed saturation of *V_spec_* at higher pectin concentrations, which deviates from traditional Michaelis–Menten kinetics (Model 1, Equation (4) in Section 3.7). Instead, a well-expressed inhibition effect occurred, with *V_spec_* decreasing above 1.0–1.2 mg/mL of pectin. This substrate inhibition was previously observed for free pectinase-mediated pectin degradation by Gummadi and Panda [31], who demonstrated that the process can be most adequately described by two models: (i) the conventional substrate-inhibition model (Model 2, Equation (5)) derived by Haldane [32], who proposed the formation of multiple inactive enzyme-substrate complexes, and (ii) the model proposed by Edwards [33], which combines diffusion-controlled substrate supply with substrate inhibition effects (Model 3, Equation (6)).

Figure 6b–f depicts the fits of Models 2 and 3 to the experimental kinetic curves for both the free enzyme and the most representative Pec@PA pectinolytic conjugates.

To ensure that fitting the experimental curves to Equations (5) and (6) will render *V_max_*, *K_m_*, and *K_i_* values with meaningful physical interpretation and reliable statistics, the procedure of Cleland for a single inhibitory substrate was followed [34]. First, kinetic data were plotted in double-reciprocal form (1/*V_spec_* vs. 1/S), and the linear portions of the plots were used to estimate the apparent *V_max_* and *K_m_* values, as in standard Michaelis–Menten kinetics (Equation (4)). The missing *K_i_* constant was initially estimated using the following approximate relationship [34]:(1)$$K_{i}=S_{max}^{2}/K_{m}$$
where *S_max_* is the substrate concentration at maximum *V_spec_* values. These starting *V_max_*, *K_m_*, and *K_i_* values were then input into a nonlinear fitting program based on Equations (5) and (6). The best fits for the most representative biocatalysts are presented in Figure 6b–f, with the corresponding *V_max_*, *K_m_*, and *K_i_* for all systems listed in Table 2.

To decide which substrate inhibition method—conventional substrate inhibition (Model 2, Section 3.7) or the double exponential model (Model 3)—best describes the kinetics of the systems presented in Table 2 and Figure 6, we first examined the coefficient of determination, *R*^2^. It can be observed that Model 3 better captures the trend of the data points for the free enzyme, Pec@PA4, and Pec@PA6, suggesting that these biocatalysts exhibit a combination of diffusion-controlled substrate supply and substrate inhibition effects. In contrast, Model 2, not accounting for diffusion, provides a more suitable fit for Pec@PA6-Fe, whereas both Model 2 and Model 3 are equally adequate for the remaining four complexes, yielding similar *V_max_*, *K_m_*, and *K_i_* values. This finding is likely due to the distinct morphology of the Pec@PA6 conjugates, as detailed in Section 2.2.

In enzymatic biocatalysis, the constant *K_i_* is a significant indicator of the substrate inhibition phenomenon, reflecting the affinity between the enzyme and the inhibitory form of the substrate. A lower *K_i_* value indicates a higher tendency for inhibitory binding, meaning that inhibition occurs even at lower substrate concentrations. Conversely, a higher *K_i_* suggests that significant inhibition only occurs at high substrate concentrations. As shown in Table 2, the Pec@PA612 conjugate, with *K_i_* = 7.5, is notably more resistant to inhibition compared to the free Pec reference, which has *K_i_* = 5.7. On the other end of the range, the Pec@PA4-Fe and Pec@PA6 conjugates are more easily inhibited, with *K_i_* values ranging from 2.2 to 2.6. The *K_i_* values for the remaining samples fluctuate between 3.5 and 4.8, which is close to, but still below, the value for the free enzyme.

Key parameters characterizing an enzymatic process include the specific maximum catalytic activity (*V_max_*) and the Michaelis–Menten constant (*K_m_*). *V_max_* represents the maximum reaction rate of the enzyme system, while *K_m_* indicates the enzyme’s substrate affinity, with a lower *K_m_* signifying higher affinity. Additionally, catalytic efficiency (CE) can be derived from these parameters. Since, in this study, *V_max_* is already normalized to the immobilized protein amount in each system, the following relationship can be expressed:(2)CE=VmaxKm

The CE value reflects the enzyme’s performance, with high velocity and a low *K_m_* indicating high catalytic efficiency. This means the catalyst is both fast and effective at binding to the substrate.

With these concepts in mind, Figure 7 was constructed to present the consolidated data for *CE* and *K_i_* for all biocatalysts in this study. The term “Pec” in complex designations is omitted for better readability.

Analyzing the *CE* and *K_i_* values for all catalytic systems in Figure 7, the most suitable candidates for further use in larger-scale applications are catalysts with both high *CE* and *K_i_*, i.e., combining efficient catalysis with resistance to inhibition at high substrate levels. The ideal choice, compared to the free Pec reference, is the Pec@PA12 system, which achieves the highest *CE* with a sufficiently high *Ki*. Another promising option is the Pec@PA612 conjugate, featuring the same CE as free Pec but with significantly higher *Ki*. The next best options are the PA4-based complexes, Pec@PA4 and Pec@PA4-Fe, followed by the PA6-supported systems, with and without Fe filler. The only system with notably low *CE* and *Ki* as compared to the free Pec is Pec@PA12-Fe, where the addition of Fe appears to have compromised both parameters.

In summary, the morphology and chemical structure of the polyamide MPs significantly influence the activity and robustness of noncovalently immobilized Pec against the pectin substrate. This insight could guide the design of customized biocatalytic systems for applications in the food industry, particularly for controlling pectin levels.

### 2.6. Rosé Must Clarification

The next step in this study was to use all biocatalysts for the clarification of real rosé must samples containing pectin concentrations of 2.0 mg mL^−1^, i.e., significantly above the threshold of 1.2 mg mL^−1^, above which the inhibition effect was registered in the kinetics studies. To ensure the comparability of the results, the amounts of the pectinolytic preparation immobilized on all seven supports were kept identical to those of the free enzyme, as expressed in arbitrary units, by the appropriate selection of the mass of the Pec@PA conjugate. The results of all clarification studies, performed at 23 ± 1 °C without stirring, are presented in Figure 8.

As shown in Figure 8a, without enzyme treatment, including the presence of a neat PA6 MP carrier, the initial turbidity remained constant at approximately 70 NTU over time. During the first hour of clarification, all immobilized complexes in Figure 8a demonstrated faster clarification than the free Pec, which reached the target turbidity of 20 NTU within 2 h. The Pec@PA4-Fe complex followed a similar trend, while the Pec@PA6 and Pec@PA6-Fe achieved 20 NTU within 160 min. Notably, Pec@PA4 exhibited the best clarification performance, reaching the target turbidity level in just 80 min.

A comparable turbidity reduction pattern was observed for the PA12-based complexes in Figure 8b. All of them reduced turbidity from 70 to 45 NTU within 60 min, outpacing the free Pec. This initial effect was the strongest with the Pec@PA612 complex. Thereafter, the clarification rate of the immobilized systems in Figure 8b slowed slightly, achieving the desired clarity between 160 and 180 min.

The results in Figure 8 align with the kinetic data, indicating that Pec@PA4 and Pec@PA12 serve as faster catalysts than the free enzyme. However, under real clarification conditions, factors beyond kinetics appear to influence performance. The pulverulent catalytic complexes—depending on the polyamide’s nature, density, and molecular weight—float on the must surface, disperse within it, or rapidly precipitate to the bottom of the container, all of which certainly impact the clarification rate additionally.

### 2.7. Color Retention After Clarification

Alongside transparency, color is a critical characteristic of rosé and white wines, subject to rigorous control. Color is typically analyzed across three bands in the UV/VIS spectrum: 420 nm (yellow), 520 nm (red), and 620 nm (blue) [35,36]. Figure 9 presents comparative results for rosé musts treated with the free enzyme and the seven immobilized conjugates.

Theoretically, immobilized enzymes could introduce undesirable color changes due to possible selective adsorption of wine color components onto the microparticulate PA supports, an effect not present with the free enzyme preparation. However, the UV/VIS curves in Figure 9a and the bar graphs in Figure 9b show that the absorbances at 420 nm, 520 nm, and 620 nm for immobilized conjugates are statistically comparable to, or even better than those of the commercial free enzyme. Figure 9c illustrates the color intensity *I* and shade *N* as functions of the Pec@PA type, calculated as per Equations (7) and (8), indicating once again that color changes caused by the free Pec are equivalent to those caused by the Pec@PA conjugates.

In summary, the microparticulate polyamide supports in all noncovalently immobilized conjugates do not induce any undesirable color changes in the rosé must, despite potentially adsorbing some pectin degradation products, as suggested by observations regarding the multiple uses of the seven conjugates.

### 2.8. Multiple Uses of Pec@PA Conjugates

The primary advantage of the MP-supported pectinolytic Pec@PA catalysts over the free Pec resides in the ability to completely remove and reuse the immobilized biocatalysts. The reusability of the Pec@PA systems was evaluated by repeatedly using defined amounts of each biocatalyst for rosé must clarification at 23 ± 1 °C without stirring, followed by enzymatic activity tests, as described in Section 3.10.

Figure 10 illustrates the results of rosé must clarification using immobilized conjugates over three consecutive applications. During the first use (red points and black squares in Figure 10a,c; pink triangles in Figure 10b), all seven complexes reduced turbidity to below 20 NTUs within approximately 3 h. However, during the second use (blue and green triangles in Figure 10a,c; brown circles in Figure 10b), achieving the same turbidity level required about 14 h.

The third application was performed using Pec@PA4 (yellow triangles and purple diamonds, Figure 10a) and Pec@PA6 (yellow triangles and purple diamonds, Figure 10b) complexes, which took 32 h to reduce turbidity to 20 NTUs. It was not possible to conduct a third cycle with Pec@PA12 complexes, as their morphology and low density, along with the lack of magnetic properties, prevented recovery after the second cycle.

The reduced relative activity during the second and third cycles most probably results from byproduct adsorption on the catalytic complex surface, which hinders substrate transport to the enzyme’s active center, prolonging the clarification process. At the industrial scale, clarification times of 14 or 32 h are still considered acceptable.

The reusability of the immobilized enzyme was also analyzed by performing consecutive cycles of enzymatic activity, as described in Section 3.10. The assays were carried out at 50 °C and pH 3.8, and the results of residual activity after each cycle are shown in Figure 11. As illustrated in Figure 11, after the first application cycle, the three Fe-containing systems (Pec@PA4-Fe, Pec@PA6-Fe, and Pec@PA12-Fe) retained 66–70% of their initial activity, while the non-Fe systems stabilized around 50–55%. This higher activity retention probably results from the easier and more complete removal of the Fe-containing systems using a constant magnet, which minimizes catalyst loss and indirectly supports higher relative activity. The three magnetically susceptible systems maintain activity above or around 50% through the seventh cycle, indicating good potential for large-scale reuse. Among the non-Fe systems, Pec@PA6 shows the best performance. In all Pec@PA conjugates, possible causes of activity decline may also include enzyme leaching and/or surface contamination of the support particles by pectin degradation products. After the final reusability cycle, the used Pec@PA particles can be boiled in distilled water to remove all accumulated contaminants, dried, and reused for a new immobilization.

### 2.9. Storage Stability of Pec@PA Conjugates

Figure 12 illustrates the storage stability of all biocatalysts in this study at 4 °C for one month, determined as detailed in Section 3.11. By the end of the test, all immobilized catalysts, except for Pec@PA612 and Pec@PA12, show stability comparable to that of the free enzyme. Notably, Pec@PA4 and Pec@PA4-Fe exhibit higher residual activity than the free enzyme up to the third week of storage, suggesting that the support may offer some enzyme protection.

Based on the results in Figure 12, it is recommended to use the Pec@PA pectinolytic systems within 2–3 weeks after immobilization to preserve their enhanced activity as compared to free Pec.

## 3. Materials and Methods

### 3.1. Reagents

The monomers 2-pyrrolidone (2PD, *purum* > 98%) and laurolactam (LL, *purum* > 99%) were purchased from Sigma Aldrich–Merck (Lisbon, Portugal) and used as received. The ε-caprolactam (ECL) monomer for anionic polymerization (AP-NylonR) and the Brüggolen C20^®^ (C20) polymerization activator were both obtained from Bruggemann Chemical (Heilbronn, Germany). According to the manufacturer, C20 contains 80 wt.% of aliphatic diisocyanate blocked in ECL. The initiator, sodium dicaprolactamato-bis-(2-methoxyethoxo)-aluminate (80 wt.% in toluene) (Dilactamate^®^, DL), was purchased from Katchem (Prague, Czech Republic) and used without further purification.

The soft, non-insulated iron particles (Fe content > 99.8%), with average diameters of 3–5 μm, were generously donated by BASF (Ludwigshafen, Germany). Viazym Clarif Extrem^®^, an enological pectinase preparation derived from *Aspergillus niger*, was sourced from Martin Vialatte (Magenta, France). This preparation is typically used for the industrial clarification of white wine and rosé musts, such as the sample quantities of industrial rosé must, which were kindly donated by Sogrape Vinhos SA (Avintes, Portugal).

All other chemical reagents, such as pectin from citrus peel, and solvents used in this study, were of analytical grade and supplied by Sigma Aldrich–Merck (Lisbon, Portugal).

### 3.2. Characterization Methods

The scanning electron microscopy (SEM) studies were carried out using a NanoSEM-200 apparatus from FEI Nova (Hillsboro, OR, USA) with mixed secondary electron/backscattered electron in-lens detection. All the samples were observed after sputter coating with an Au/Pd alloy using a 208 HR instrument from Cressington Scientific Instruments (Watford, UK), featuring high-resolution thickness control. The UV-VIS spectral measurements were performed with a Shimadzu model 1900i double-beam spectrophotometer (Tokyo, Japan) working in photometric or spectral modes. Fourier-transform infrared spectroscopy with attenuated total reflection (FTIR-ATR) was conducted, and the spectra were collected using a Perkin-Elmer Spectrum 100 apparatus (Waltham, MA, USA) with a horizontal ATR attachment featuring a ZnSe crystal. Spectra were acquired between 4000 and 600 cm^−1^, accumulating up to 16 spectra with a resolution of 2 cm^−1^. The samples were studied in the form of fine powders. The turbidity measurements were made using a METRIA M10 portable turbidimeter (Labbox Labware, Barcelona, Spain), performing up to 5 parallel measurements with every sample, according to [37].

### 3.3. Synthesis of PA6 Microparticles

The PA6, PA12, and the copolymeric PA612 MPs were prepared by AAROP in suspension, following the general method described by Dencheva et al. [26]. Polymerizations were conducted in a 250 mL flask equipped with a thermometer, magnetic stirrer, Dean-Stark attachment for azeotropic distillation with a reflux condenser, and a nitrogen inlet. The monomers (0.3 mol of ECL for PA6 MP, 0.2 mol of LL for PA12 MP, and a mixture of 0.1 mol CL and 0.1 mol LL for PA612 MP) were added to 100 mL of a mixed hydrocarbon solvent (toluene/xylene, 1:1 by volume) under a nitrogen atmosphere, and the mixture was refluxed for 10–15 min. For Fe-containing MPs, 1–3 wt.% of iron microparticles (relative to monomer) were introduced at this point. Then, 3.0 mol% of DL and 1.5 mol% of C20 were added simultaneously. The reaction proceeded for 2 h after the catalyst addition, maintaining the temperature between 125–135 °C for PA6 synthesis or around 120 °C for PA12 and PA612, with constant stirring at 800 rpm. The fine powders produced were isolated by dissolving the reaction mixture in 80 mL of acetone to remove unpolymerized monomer, followed by vacuum filtration and washing with 100 mL of methanol to eliminate low molecular weight oligomers. The resulting microparticles were dried for 2 h in a vacuum oven at 80 °C. All MP powders were subjected to Soxhlet extraction with methanol for 4 h to remove traces of oligomers or impurities, then dried and stored in a desiccator.

For the PA4-Fe MPs, low-temperature solventless AAROP was employed [38]. In this case, 0.2 mol of 2PD was stirred with 1 wt.% of Fe at 25 °C for 30 min. C20 activator and DL initiator (1.5 mol % and 3.0 mol %, respectively) were added under stirring in an inert atmosphere at 20 °C. The temperature was then raised to 40 °C, and the pressure was reduced to 50 mbar for 6 h. After completion, the reaction mixture was dispersed in acetone, filtered, and washed twice with methanol to remove unreacted monomer. Oligomers were removed by Soxhlet extraction with methanol for 4 h, followed by drying in a vacuum and storage in a desiccator. The procedure for neat PA4 MP followed the same steps, without the addition of magnetic fillers in the monomer feed.

### 3.4. Immobilization of Pectinase by Physical Absorption

The commercial enological pectinase preparation, Viazym Clarif Extrem^®^, was immobilized by adsorption on the seven different microparticulate supports in the following way. The PA microparticles (100 mg) were incubated with 5 mL of the pectinase preparation diluted 10-fold in double-distilled water, under gentle agitation, at 23 ± 1 °C. It was assumed that the commercial pectinase preparation contained 10 a.u. of protein per mL. After 24 h, all samples were centrifuged, and the supernatant was decanted and stored at 4 °C for further analyses. The final Pec@PA complexes were washed twice with double-distilled water to remove any non-immobilized enzymes and stored at 4 °C for subsequent use. Furthermore, a series of experiments were conducted to gather the data necessary to define an immobilization process, as suggested in [39].

### 3.5. Determination of the Total Amount of Protein

After immobilization, the supernatants were subjected to UV analysis to determine the residual protein content and calculate the total protein (TP) incorporated into the PA microparticles (MPs), expressed as:(3)$$TP=C_{0}-C_{s},\;\left(a.u.\right)$$
where *C*_0_ is the initial protein content before immobilization, and *C_s_* is the protein content in the supernatant after the immobilization process. TP was determined by direct quantification of the UV absorption peak at λ ≈ 263 nm. The absorbance was measured, and *C_s_* was determined using a standard calibration curve. In this study, TP is considered as the immobilization yield.

### 3.6. Pectinase Activity Assay

Pectinase activity was determined by measuring the amount of reducing sugars released from a pectin solution using the DNS method proposed by Miller [30]. The DNS reagent consists of 2-hydroxy-3,5-dinitrobenzoic acid (1% *w*/*v*), potassium sodium tartrate (30% *w*/*v*), and sodium hydroxide (1% *w*/*v*). Citric pectin (0.25% *w*/*v*) in citrate buffer (50 mM, pH 3.8) was used as the substrate. Typically, free pectinase preparation (0.1 mL, 0.1 a.u. total protein) or immobilized enzyme (0.020 g) was mixed with the substrate solution (0.90 mL) and incubated at 50 °C for 15 min. The reaction mixture (1 mL) was then combined with DNS solution (1 mL), and the samples were kept in boiling water for 10 min before being cooled on ice to stop the reaction. Absorbance was measured at 540 nm against the blank DNS reagent. The amount of released reducing sugar was quantified using a D-galacturonic acid standard calibration plot in the concentration range of 0.28–2.83 µmol/mL. One unit of pectinase activity is defined as the amount of enzyme required to release 1 μmol of galacturonic acid per minute under standard assay conditions.

### 3.7. Kinetics of Pectin Degradation

The kinetic parameters of pectin degradation by free pectinase and the immobilized Pec@PA complexes were determined by measuring the degradation of pectin to D-galacturonic acid at various substrate concentrations (0.05–5 mg/mL), with an incubation time of 1 min under the conditions previously described in Section 3.6. The total amount of enzyme (0.15 a.u.) added to the reaction mixture was the same in all cases. Three kinetic models, based on previous studies, were selected to calculate the kinetic parameters, namely the Michaelis–Menten model and two models accounting for substrate inhibition. The equations for these models are presented below: Equations (4)–(6) [31]:(4)$$V=\frac{V_{max}S}{K_{m}+S}\left(\mathrm{Model}\;1,\;\mathrm{Michaelis}–\mathrm{Menten}\;\mathrm{equation}\right)$$
(5)$$V=\frac{V_{max}S}{\left(K_{m}+S\right)\left(1+\frac{S}{K_{i}}\right)}\left(\mathrm{Model}\;2,\;\mathrm{conventional}\;\mathrm{substrate}\;\mathrm{inhibition}\;\mathrm{model}\right)$$
(6)$$V=V_{max}\left[\exp\left(\frac{-S}{K_{i}}\right)-\exp\left(\frac{-S}{K_{m}}\right)\right]\;\;\;\left(\mathrm{Model}\;3,\;\mathrm{double}\;\mathrm{exponential}\;\mathrm{model}\right)$$

The kinetic parameters, including the maximal reaction rate ($V_{max}$), Michaelis–Menten constant ($K_{m}$), and the constant characterizing the formation of an inactive enzyme–substrate complex ($K_{i}$)), were computed using nonlinear multiple regression analysis with the commercial software package OriginPro, version 9.8.0.200 by OriginLab Corporation (Northampton, MA, USA).

### 3.8. Application of Immobilized Pectinase for Rosé Must Clarification

Clarification of the rosé must was carried out at 23 ± 1 °C, without stirring, using both free and immobilized pectinase preparations. The manufacturer’s recommended concentration of free enzymatic preparation and complexes containing the same amount of immobilized enzyme were used. The enzyme was mixed with 14 mL of rosé must for three hours, and turbidity was measured in nephelometric turbidity units (NTUs) every 20 min. A pectin test was then performed to detect the presence of residual pectin by adding acidified alcohol (5% *v*/*v* HCl, 5 mL) to 2.5 mL of the must sample under gentle agitation. After 10 min, the must was visually inspected. In the presence of residual pectin, insoluble flakes form; if the liquid is clear, the sample contains no residual pectin.

### 3.9. Color Analyses

The color of the must was analyzed by UV/VIS spectroscopy (method OIV-MA-AS2-07B) [35], to evaluate any color changes resulting from enzymatic clarification. Spectra were obtained from all treated samples both before and after clarification. The color of the must was measured at three wavelengths: 420, 520, and 620 nm.

The color intensity (*I*) was calculated using Equation (7). The darkness or lightness of the color, known as the shade (*N*), was determined using Equation (8).
(7)$$I=A_{420}+A_{520}+A_{620}$$
(8)N=A420A520
where $A_{420},\;A_{520},$ and $A_{620}$ represent the absorbance values at 420 nm, 520 nm, and 620 nm, respectively.

### 3.10. Reusability Studies

The Pec@PA conjugates, each containing 0.15 a.u. of enzyme, were used for must clarification over three consecutive cycles. Turbidity was measured until levels of around 20 NTUs were reached, as described in Section 3.8. After the first clarification cycle, the complexes were recovered from the must by centrifugation, washed twice with 1 mL of deionized water, and tested for their clarification capacity with fresh rosé must, following the same procedure.

For enzymatic activity tests in this study, Pec@PA complexes (0.020 g) were mixed with 0.90 mL of citric pectin substrate, prepared as previously described in Section 3.6, and incubated at 50 °C. After a 15 min reaction, the conjugates’ microparticles were washed twice with deionized water to remove any residual substrate, and their activity was tested with fresh substrate. This process was repeated for eight consecutive cycles. Activity after each cycle was measured in terms of residual activity, with the first cycle set at 100%.

### 3.11. Storage Stability Studies

Free pectinase and immobilized enzyme complexes Pec@PA were stored at 4 °C for one month, the latter in double-distilled water. Every week, enzyme samples were taken, and their residual enzymatic activity was measured by the DNS method, as previously described. The activity of the first week was considered 100%.

## 4. Conclusions

This study presents the first systematic investigation of the influence of polyamide support type on the properties of catalytic complexes produced via non-covalent immobilization of a commercial pectinase preparation. Seven distinct supports are synthesized by AAROP, differing in chemical structure, composition, shape, size, and surface topology of the microparticles. The immobilization process employs a simple, one-pot method: incubating the supports in buffered aqueous enzyme solutions without the use of hazardous chemicals, thereby ensuring compatibility with food industry requirements.

The results demonstrate a significant effect of polyamide type (PA4, PA6, PA12, with and without Fe carrier, and copolymeric PA612) on the enzyme immobilization yield and the activity of the resulting Pec@PA catalytic conjugates. Comparative kinetic studies revealed that, except for Pec@PA12-Fe, all conjugates exhibited higher catalytic efficiency than free pectinase. This finding should be ascribed to the different structure and morphology of the polyamide supports.

The polyamide-immobilized pectinase biocatalysts in this study achieved effective rosé must clarification at the laboratory scale within an industrially relevant time frame of 80–180 min during the first clarification round. Most immobilized complexes were reusable for second and third consecutive clarification cycles, with clarification times extending to approximately 15 and 30 h, respectively, which is still within industrially acceptable limits. Additionally, each conjugate supported up to seven consecutive clarification rounds, with most systems retaining about 50% of their initial activity.

Notably, the polyamide supports in the Pec@PA conjugates did not alter the must color, as confirmed by UV/VIS measurements across three key wavelength ranges. Furthermore, all immobilized systems, except for Pec@PA12, retained up to 75% of their initial activity after four weeks of storage. These properties are comparable to those of the free pectinase preparation.

Currently, enological assessments are underway to evaluate the sensory properties of the clarified musts using the best-performing polyamide-immobilized pectinase conjugates. These studies will provide a foundation for practical applications of these novel biocatalysts.

## Figures and Tables

**Figure 1 molecules-30-00114-f001:**
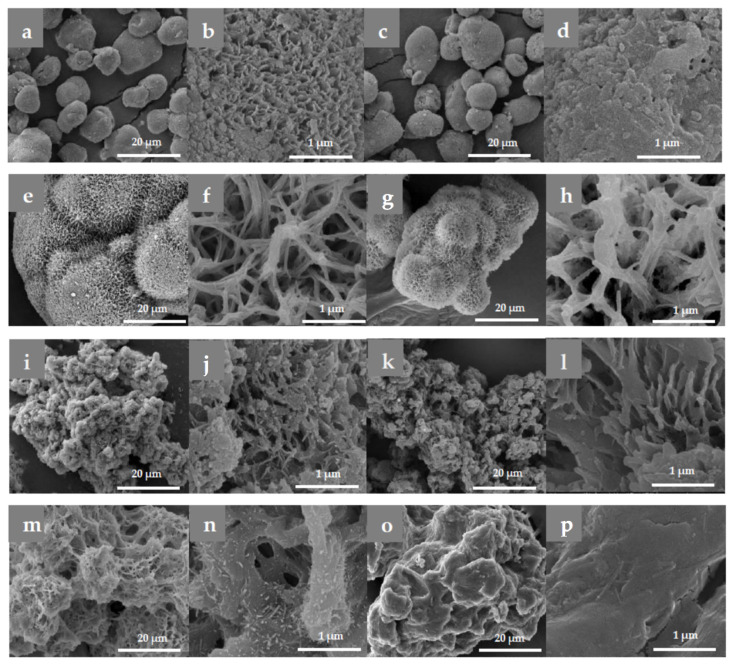
Selected SEM micrographs of empty supports and pectine-immobilized conjugates: (**a**,**b**) PA4; (**c**,**d**) Pec@PA4; (**e**,**f**) PA6; (**g**,**h**) Pec@PA6; (**i**,**j**) PA12; (**k**,**l**) Pec@PA12-Fe; (**m**,**n**) PA612; (**o**,**p**) Pec@PA612 complexes.

**Figure 2 molecules-30-00114-f002:**
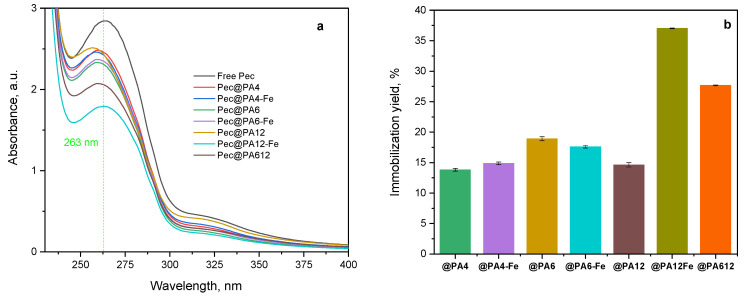
Quantification of the pectinase immobilization: (**a**) UV/VIS spectra of the Viazym solutions before and after adsorption immobilization on seven different MP supports; (**b**) immobilization yield for all Pec@PA and Pec@PA-Fe pectinolytic complexes. For more data, see the text.

**Figure 3 molecules-30-00114-f003:**
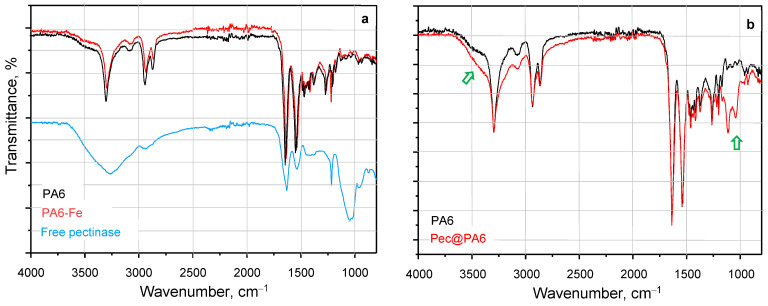
Selected FTIR spectra of PA6 supports, before (**a**) and after (**b**) noncovalent Pec immobilization. The two arrows in (**b**) indicate the Pec bands present in the Pec@PA6 spectra.

**Figure 4 molecules-30-00114-f004:**
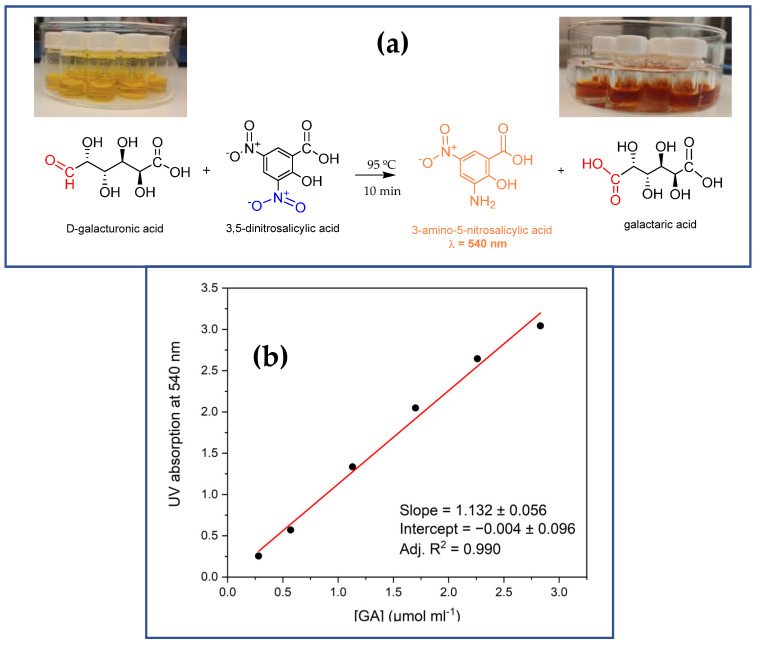
Activity assay of pectinolytic biocatalysts by the DNS method. (**a**) reaction scheme of the interaction between 3,5-DNS and D-galacturonic acid (GA); (**b**) the standard calibration curve constructed with various concentrations of GA.

**Figure 5 molecules-30-00114-f005:**
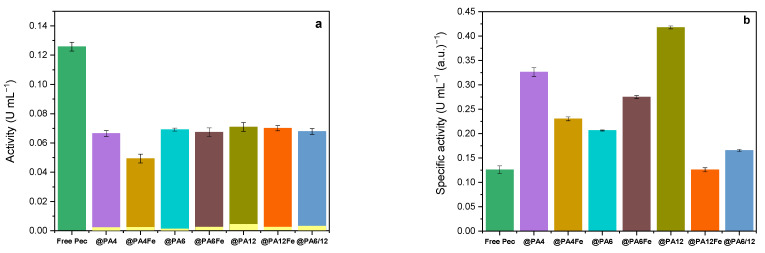
Activity of Pec@PA immobilized conjugates in comparison to free Pec: (**a**) absolute activity. The small yellow bars in **a**) show the “activity” values of the respective empty PA supports.; (**b**) specific activity per unit enzyme (free or immobilized).

**Figure 6 molecules-30-00114-f006:**
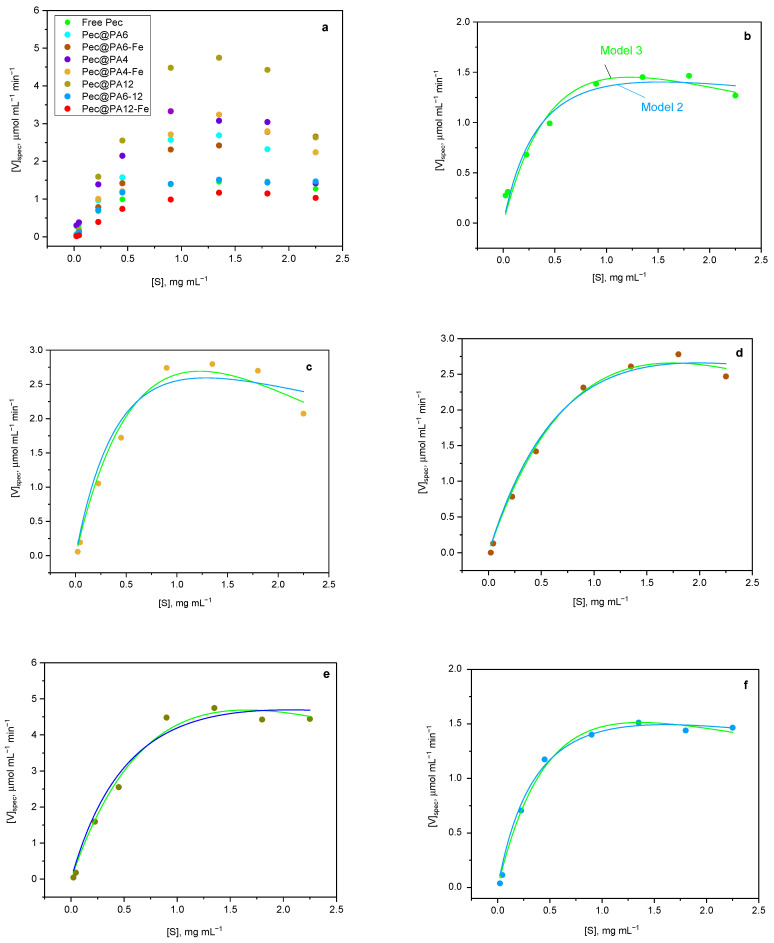
Selected kinetic curves and their best fits with Model 2 (blue curves) and Model 3 (green curves) for: (**a**) comparison of all experimental curves; (**b**) free pectinase; (**c**) Pec@PA4-Fe; (**d**) Pec@PA6-Fe; (**e**) Pec@PA12; (**f**) Pec@PA612.

**Figure 7 molecules-30-00114-f007:**
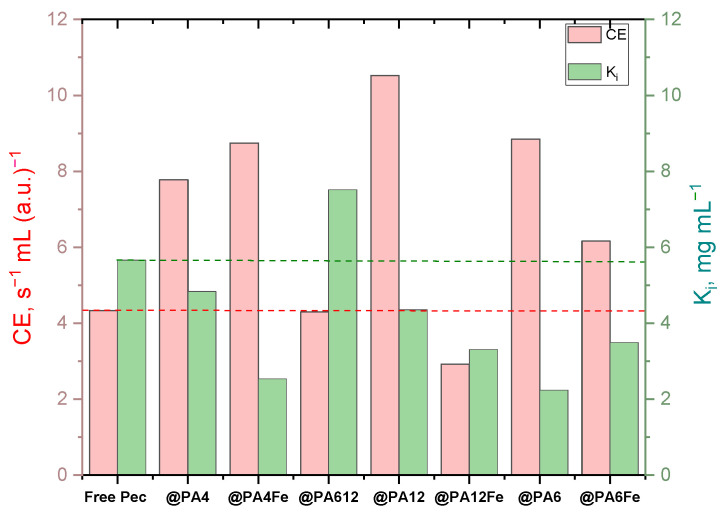
Consolidated results from the kinetics studies. *CE =* catalyst efficiency, *K_i_* = inhibition constant. The dashed lines are guides for the eye, highlighting the data of the free enzyme reference.

**Figure 8 molecules-30-00114-f008:**
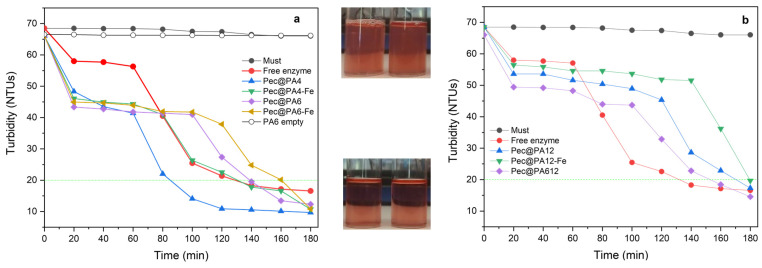
Clarification studies with free and immobilized pectinase preparations: (**a**) free Pec compared to PA4- and PA6-based Pec@PA complexes; (**b**) free Pec compared to PA12-based Pec@PA complexes. All immobilized catalysts contain 0.15 a.u. of pectinolytic enzyme; the same amount is present in the free Pec catalyst. The dashed lines indicate the level of 20 NTUs, below which the must is considered completely clarified, i.e., not containing pectin. The visual aspects of the must before and after clarification are presented in the upper and lower photographs, respectively.

**Figure 9 molecules-30-00114-f009:**
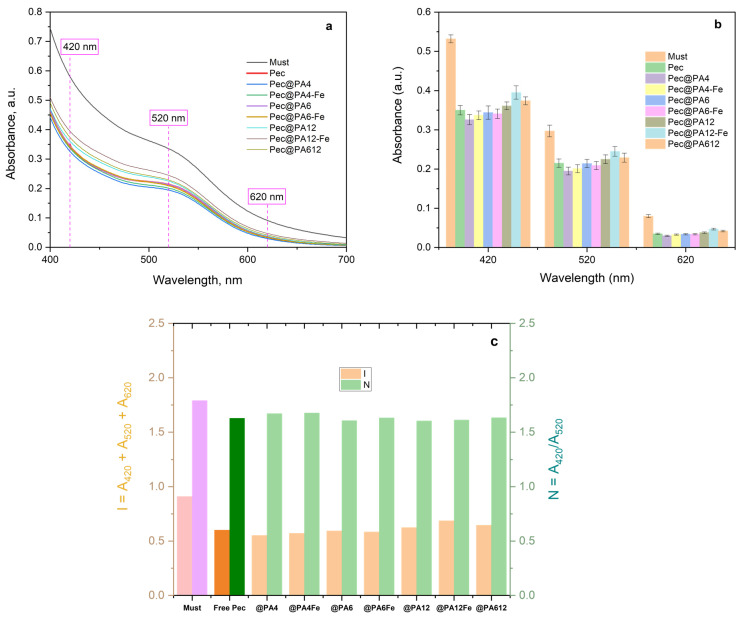
Color changes in treated rosé must samples: (**a**) UV/VIS spectra; (**b**) data at 420, 520, and 620 nm; (**c**) *I* and *N* color parameters before and after treatment with various pectinolytic catalysts. “Pec” in most sample labels is omitted for better readability. Bar colors in (**c**) distinguish untreated must and that with free Pec treatment from all other samples. For more details, see Section 3.9.

**Figure 10 molecules-30-00114-f010:**
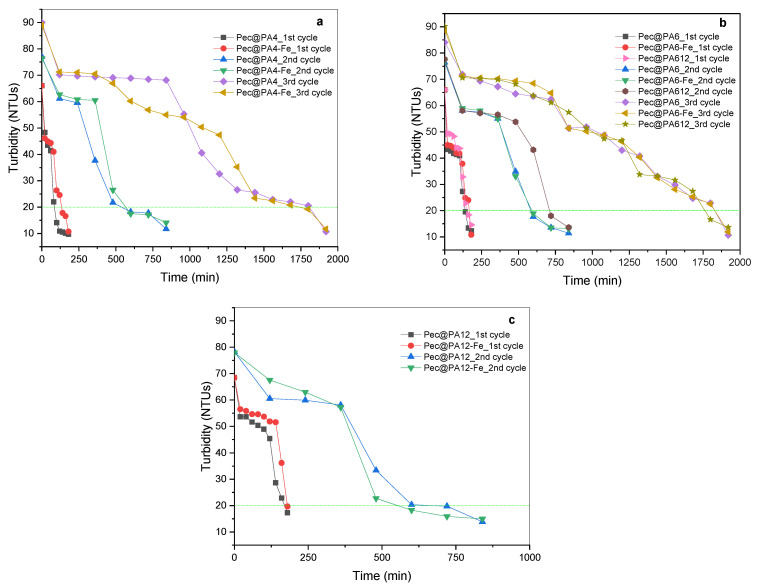
Clarification studies of rosé must with Pec@PA systems in several different clarification cycles: (**a**) with Pec@PA4 and Pec@PA4-Fe; (**b**) with Pec@PA6, Pec@PA6-Fe, and Pec@PA612; (**c**) with Pec@PA12 and Pec@PA12-Fe. All immobilized catalysts contain 0.15 a.u. of the pectinolytic enzyme. The dashed lines indicate the level of 20 NTUs, below which the must is completely clarified.

**Figure 11 molecules-30-00114-f011:**
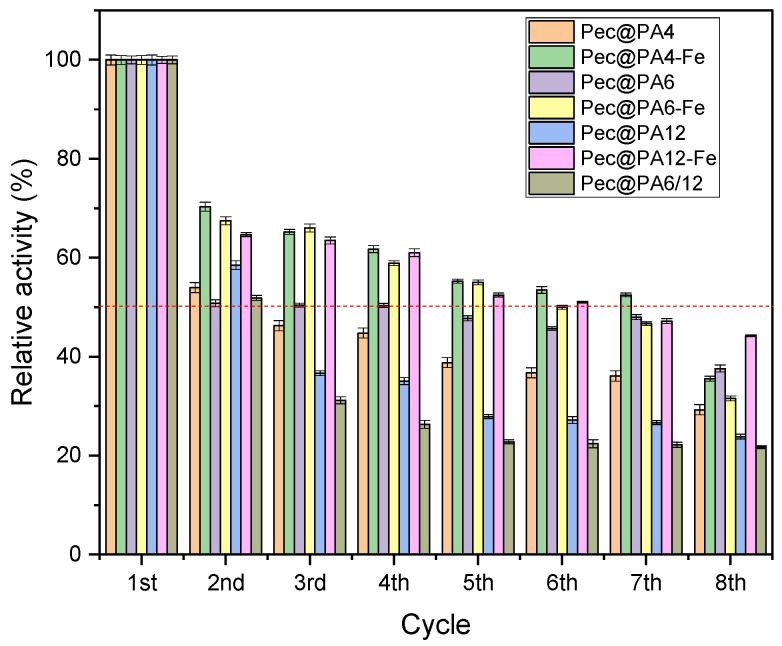
Relative activity of immobilized Pec@PA pectinolytic catalysts over eight application cycles. The red dashed line indicates 50% relative activity. Initial activity (Figure 5b) is set to 100% for each complex. See text for details.

**Figure 12 molecules-30-00114-f012:**
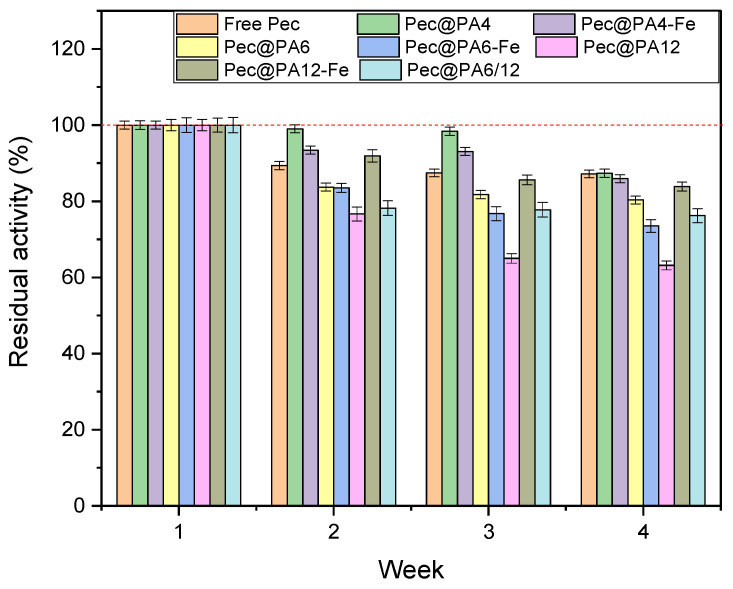
Storage stability of immobilized Pec@PA pectinolytic catalysts over four weeks at 4 °C. Initial activity (Figure 5b) is normalized to 100% for each complex.

**Table 1 molecules-30-00114-t001:** Some basic characteristics of the synthesized microparticulate polyamide supports.

Sample	Yield of Polymer, %	Oligomers ^1^, wt.%	*M_η_* kDa ^2^	Average Particle Size, µm	Average Pore Size, nm	Fe Content ^4^, wt.%
PA4	59.4	4.4	26.4	10–25	50–120	-
PA4-Fe	61.2	5.9	- ^3^	10–40	120–250	1.0 [1.6]
PA6	56.0	2.0	33.7	50–150	300–500	-
PA6-Fe	58.8	2.4	-	50–200	200–300	3.0 [5.6]
PA12	42.0	3.2	33.8	50–700	250–900	-
PA12-Fe	58.5	3.4	-	50–80	750–1500	2.0 [4.2]
PA612	77.0	2.1	28.9	100–150	250–2000	-

^1^ After Soxhlet extraction; ^2^ Viscosity-average molecular weight M_η_ determined in H_2_SO_4_; ^3^ Formation of precipitate, intrinsic viscosity η impossible to determine; ^4^ In relation to monomer before polymerization and in the final polymer (in brackets).

**Table 2 molecules-30-00114-t002:** Kinetic parameters estimated for Pec and the seven Pec@PA catalytic conjugates.

Biocatalyst	Parameters ^(a)^	Model 1	Model 2	Model 3
Free Pec	*V_max_* *K_m_* *K_i_*	1.956 ± 0.024	2.182 ± 0.071	1.933 ± 0.001
		0.423 ± 0.009	0.423 ± 0.043	0.446 ± 0.041
		≈3.0	5.520 ± 0.850	5.656 ± 0.687
	*Adj. R* ^2^	0.995	0.958	**0.997**
Pec@PA4	*V_max_*	5.672 ± 0.020	9.364 ± 0.014	5.592 ± 0.101
	*K_m_*	0.752 ± 0.010	1.444 ± 0.035	0.720 ± 0.020
	*K_i_*	2.5–3.0	1.920 ± 0.060	4.838 ± 0.304
	*Adj. R* ^2^	0.989	0.965	**0.994**
Pec@PA4-Fe	*V_max_*	4.193 ± 0.027	5.387 ± 0.015	5.981 ± 0.164
	*K_m_*	0.662 ± 0.012	0.695 ± 0.012	0.684 ± 0.027
	*K_i_*	2.0–3.0	2.390 ± 0.025	2.536 ± 0.100
	*Adj. R* ^2^	0.977	0.981	0.985
Pec@PA6	*V_max_*	4.669 ± 0.020	5.568 ± 0.005	7.679 ± 0.170
	*K_m_*	0.863 ± 0.009	0.887 ± 0.002	0.868 ± 0.050
	*K_i_*	≈2.5	2.359 ± 0.021	2.236 ± 0.131
	*Adj. R* ^2^	0.995	0.988	**0.995**
Pec@PA6-Fe	*V_max_*	5.100 ± 0.020	6.136 ± 0.003	6.133 ± 0.644
	*K_m_*	1.222 ± 0.008	1.297 ± 0.002	0.995 ± 0.020
	*K_i_*	≈2.5–3.0	3.035 ± 0.060	3.491 ± 0.554
	*Adj. R* ^2^	0.995	**0.998**	0.984
Pec@PA12	*V_max_*	8.666 ± 0.019	8.335 ± 0.016	8.408 ± 0.196
	*K_m_*	1.010 ± 0.007	0.869 ± 0.032	0.799 ± 0.030
	*K_i_*	≈2.5	5.323 ± 0.066	4.354 ± 0.226
	*Adj. R* ^2^	0.988	0.990	0.990
Pec@PA12-Fe	*V_max_*	1.872 ± 0.070	2.217 ± 0.040	2.395 ± 0.040
	*K_m_*	0.820 ± 0.020	0.855 ± 0.030	0.821 ± 0.019
	*K_i_*	≈2.5	3.737 ± 0.021	3.308 ± 0.167
	*Adj. R* ^2^	0.976	0.993	0.994
Pec@PA612	*V_max_*	1.931 ± 0.043	2.254 ± 0.020	1.924 ± 0.020
	*K_m_*	0.372 ± 0.022	0.454 ± 0.010	0.448 ± 0.020
	*K_i_*	≈5.0	5.708 ± 0.083	7.516 ± 0.539
	*Adj. R* ^2^	0.966	0.990	0.993

^(a)^ Estimations for *V_max_* (µmol mL^−1^ min^−1^) and *K_m_* (mg mL^−1^) were obtained from the linear part of the double reciprocal Lineweaver-Burk plot (Model 1, Equation (4), Section 3.7). *K_i_*, (mg mL^−1^), was estimated from Equation (1). The final *V_max_*, *K_m_*, and *K_i_* were determined after fitting of the estimates to Models 2 and 3. The bolded *R*^2^ values correspond to the better fitting model.

## Data Availability

All the data generated during this research are presented in the manuscript and in the Appendix A.

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
