# Peer review of "Polyamide Microparticles with Immobilized Enological Pectinase as Efficient Biocatalysts for Wine Clarification: The Role of the Polymer Support"

_molecules, 2024, doi:10.3390/molecules30010114_

Round 1

Reviewer 1 Report

Comments and Suggestions for Authors

In my opinion, the manuscript can fit the scope of the journal. Enzyme immobilisation is a valuable topic for sustainable development. Moreover, the English is clear, and self-citations are fine. The manuscript is well written; I made a few comments in the Introduction, Results and Discussion and Materials and Methods. My main recommendation is to underline in the conclusion the novelty of this manuscript with respect to the literature and the previous work of the authors.

In the Keywords, I would suggest adding “biocatalyst reusability”.

Line 67: I suggest referencing a few examples of immobilised enzyme systems for biocatalytic reactions to support the sentence and show that this method is used in sectors (such as fine chemicals) other than the food industry. As suggestions:

https://doi.org/10.1016/j.tetasy.2015.04.009 and  https://doi.org/10.1002/elsc.201900119

I also suggest writing that “enzyme immobilisation facilitates the development of continuous processes”.

Line 134-136: Is there any explanation of the “rapid spontaneous MP growth” in the case of PA12?

Line 247: Figure 2, please check the readability of the figures. Some words are hard to read.

Line 323: Figure 4, the small table in (b) is difficult to read.

Line 380: Figure 6: is there a reason why Figure 6(e) has a different colour of the curves than other figures?

Line 701-716: to help the reproducibility of the experiments. Is it possible to know the volume of acetone used for dissolving the reaction mixture and the volume of methanol used to wash and perform the Soxhlet extraction?

Line 808: Conclusions: what is the novelty of previous work and other literature? As written in the introduction, ”This study extends our previous approach by synthesizing other types of polyamide“. For this reason, I recommend underlining the novelty of the work in the conclusion.

Author Response

We thank the esteemed Reviewer 1 for the thorough analysis of our submission and the valuable critical notes and suggestions, which have contributed to improving our manuscript. A detailed, point-by-point answer (A) to each query (Q) is provided below.

Q1. In the Keywords, I would suggest adding “biocatalyst reusability”.

A1. Done as requested.

Q2. Line 67: I suggest referencing a few examples of immobilised enzyme systems for biocatalytic reactions to support the sentence and show that this method is used in sectors (such as fine chemicals) other than the food industry. As suggestions: https://doi.org/10.1016/j.tetasy.2015.04.009  and  https://doi.org/10.1002/elsc.201900119 I also suggest writing that “enzyme immobilisation facilitates the development of continuous processes”.

A2. We have added a phrase about continuous processes in lines 66–67, as this is an enhancement to the state-of-the-art description relevant to the article's scope.

Regarding the two additional references, they refer to specific enzymes and processes unrelated to the scope of our study. We believe these references would be appropriate for a review article, however, they are not pertinent to our focused investigation. We hope the esteemed reviewer could positively consider our reasoning.

Q3. Line 134-136: Is there any explanation of the “rapid spontaneous MP growth” in the case of PA12?

A3. The rapid spontaneous MP growth in the case of PA12 is attributed to the significantly higher reactivity of α,ω-dodecalactam during its activated anionic polymerization to PA12, compared to ε-caprolactam polymerizing to PA6 via the same process. As a result, PA12 tends to form large chunks rather than fine microparticulate powders. Lowering the temperature halts the polymerization, while increasing it promotes the formation of these chunks. Therefore, we decided to stop the process at lower degrees of conversion. An alternative solution could involve adjusting the initiator-to-activator ratio or incorporating specific additives to control the molecular weight of PA12.  These and other details are discussed in our recent article dedicated to the synthesis of PA12-based microparticles and composites on their basis https://doi.org/10.1002/app.51784.  In the original manuscript, we deliberately omitted this discussion, as it relates to optimizing polymerization processes for support preparation and lies outside the scope of this study.

Q4. Line 247: Figure 2, please check the readability of the figures. Some words are hard to read.

A4. The designations of the pectinolytic complexes in Figure 2b were really difficult to read. To address this, we shortened the designations, allowing for an increase in font size. The same changes were made also in Figs. 5a,b, 7 and 9c. We believe these modifications adequately resolve the reviewer’s concern.

Q5. Line 323: Figure 4, the small table in (b) is difficult to read.

A5. We have revised Figure 4b to display only the most important information from the linear regression analysis having increased the font size for better readability.

Q6. Line 380: Figure 6: is there a reason why Figure 6(e) has a different colour of the curves than other figures?

A6. Yes, the green curves in Fig. 6b–f represent fittings according to Model 3, while the blue curves correspond to fittings to Model 2. To better clarify this, we have added the explanation to the figure legends in Figure 6.

Q7. Line 701-716: to help the reproducibility of the experiments. Is it possible to know the volume of acetone used for dissolving the reaction mixture and the volume of methanol used to wash and perform the Soxhlet extraction?

A7. In lines 704–705 of the revised manuscript, we specified the amounts of acetone (80 mL) and methanol (100 mL) used to separate the final polymer powders. The subsequent methanol extraction was performed using a Soxhlet apparatus equipped with a 500 mL flask, filled to approximately half its volume. All acetone and methanol used in these procedures were later recycled via distillation in a rotary evaporator.

Q8. Line 808: Conclusions: what is the novelty of previous work and other literature? As written in the introduction”. This study extends our previous approach by synthesizing other types of polyamide“. For this reason, I recommend underlining the novelty of the work in the conclusion.

A8.  We appreciate this important comment. To clarify our contributions to the state of the art, we have added the phrase: “This study extends our previous approach, which was limited to PA6, by synthesizing other types of polyamides…” to line 110 of the revised manuscript. Additionally, we have revised the opening paragraph of the Conclusions section to include: “This study presents the first systematic investigation of the influence of polyamide support type on the properties of catalytic complexes produced via non-covalent immobilization of a commercial pectinase preparation. Seven distinct supports were synthesized…”

Reviewer 2 Report

Comments and Suggestions for Authors

The reviewed manuscript concerns the immobilization of pectinase in order to improve parameters related to activity and durability as well as the possibility of repeated use. This enzyme is used for wine clarification. The manuscript presents the results that are a continuation of the research conducted in the research group on the search for effective carriers for enzyme immobilization. Enzyme immobilization occurs physically by trapping the enzyme in the structure of an appropriately prepared polymer. The efficiency of enzyme immoblization is closely related to the number of gaps in the polymer. Through immobilization, the authors obtained an active preparation whose catalytic activity was improved compared to the native enzyme. Moreover, it was shown that the immobilizer can be used several times, which reduces the overall costs of the process. Improved stability  of the stored immobilizate without loss of significant activity was also noted. The planned experiments and the obtained results were correctly and carefully analyzed. In my opinion, the manuscript deserves to be published in its current form.

Author Response

We sincerely thank Reviewer 2 for the recognition and appreciation of our work.

Reviewer 3 Report

Comments and Suggestions for Authors

The present study describes the preparation and characterization of polyamide (PA) supports to be used as support materials to immobilize a commercial pectinolytic preparation (Pec). The resulting biocatalysts were utilized in wine clarification. The paper is very interesting and well-organized. However, some points need to be addressed:

- The authors should report the better results and experimental conditions for wine clarification.

- Introduction: The authors should better explore the possible mechanisms involved in the immobilization of enzyme on the resulting support, focusing on the possible mechanisms of interation.

- Introduction: Explain why this commercial pectinase formulation was chosen to perform this study.

- Introduction: How is this system different to other reports to merit publication? Please, report.

- Introduction: The end of the introduction should be a remark of the interest of the study.

- Results and discussion: Figs. 2, 5 and 10. The authors should improve the resolution of these figures. The data summrized in Axis X are hard to read.

- Results and discussion: The immobilization paramaters should be better explained. I recommend better explore the influence of the support geometry on the immobilized protein amount, catalytic activity and specific activity (explore diffusive effects on the catalytic pewrformance of the enzyme). These information is very importante to the readers.

- Materials and methods: It is a purified or a crude enzyme extract¿ Please, explain

- Materials and methods: The authors should better explain how the immobilization parameters were determined (immobilized protein amount, specific activity). See and cite this reference: https://doi.org/10.1016/j.procbio.2019.11.026.

Author Response

The present study describes the preparation and characterization of polyamide (PA) supports to be used as support materials to immobilize a commercial pectinolytic preparation (Pec). The resulting biocatalysts were utilized in wine clarification. The paper is very interesting and well-organized. However, some points need to be addressed.

We thank the esteemed Reviewer 3 for the thorough analysis of our submission and the valuable critical notes and suggestions, which have contributed to improving our manuscript. A detailed, point-by-point answer (A) to each query (Q) is provided below.

Q1. The authors should report the better results and experimental conditions for wine clarification.

A1. In the original manuscript, the conditions for wine clarification were detailed in Section 3.8 of the Materials and Methods section (lines 775-784). To address the reviewer’s comment, we have now highlighted these conditions in other parts of the manuscript as well (see lines 566 and 470–471).

Q2. Introduction: The authors should better explore the possible mechanisms involved in the immobilization of enzyme on the resulting support, focusing on the possible mechanisms of interactions.

A2. As stated in lines 98–99 of the original manuscript, the non-covalent immobilization of enzymes on polyamide microparticulate supports described in this study relies on effective multiple hydrogen bond formation between the apoenzyme peptide matrix and the amide groups of the microparticulate support. The possible mechanisms are discussed in lines 253–259 in relation to the FTIR spectra presented in Figure 3. We believe this discussion is sufficient for the scope of the present article.

More advanced evidence for these H-bond interactions, including synchrotron X-ray scattering data, was previously demonstrated in studies on laccase immobilized on PA4 and PA6 (see https://doi.org/10.3390/catal10070767). These findings are not explicitly included in the current article to maintain focus and avoid excessive self-citation.

Q3. Introduction: Explain why this commercial pectinase formulation was chosen to perform this study.

A3. As explained in lines 670-674, the Viazym Clarif Extrem used in this study is a commercial enzymatic preparation from a French manufacturer. According to its data sheet, it is a purified enological pectinase solution derived from Aspergillus Niger. This product is widely used by our industrial partner, Sogrape Vinhos SA, one of Portugal’s largest producers of white and rosé wines. Additionally, we have conducted preliminary kinetic experiments with a pectinase powder commercialized by Merck (Sigma Aldrich). However, due to its high cost, this product is not suitable for industrial applications. We plan to publish the results of these experiments separately upon their completion.

Q4. Introduction: How is this system different to other reports to merit publication? Please, report.

A4. If we understand this query correctly, the question concerns how pectinase immobilized on polyamide microparticles outperforms the free enzyme in wine clarification. This is addressed in detail at the end of the Introduction (lines 104–109) with reference to our previously published work [27] on Pec@PA6 complexes.

It is also possible that Reviewer 3 is suggesting a comparison of Pec@PA catalytic systems with pectinase immobilized on other supports, such as alginates or polyacrylamide, or even with different enzymes immobilized on various supports. We believe that such a discussion falls outside the scope of the Introduction in this focused study and would be more appropriate for a review article. We hope the esteemed Reviewer 3 will positively consider this perspective.

Q5. Introduction: The end of the introduction should be a remark of the interest of the study.

A5. Lines 110–119 in the revised manuscript outline the primary goal of this study: extending the research on Pec@PA6 complexes to five additional polyamide supports and conducting comparative analyses. We believe these additions adequately address the concern raised by Reviewer 3.

Q6. Results and discussion: Figs. 2, 5 and 10. The authors should improve the resolution of these figures. The data summarized in Axis X are hard to read.

We appreciate this important suggestion. We have made the necessary modifications to Figures 2b, 5ab, Fig. 7 and 9c to increase the font size and improve readability. Figures 10a–c, however, do not exhibit any readability issues and were therefore left unchanged.

Q7. Results and discussion: The immobilization parameters should be better explained. I recommend better explore the influence of the support geometry on the immobilized protein amount, catalytic activity and specific activity (explore diffusive effects on the catalytic performance of the enzyme). This information is very important to the readers.

A7. We appreciate this suggestion; however, the original manuscript addresses the mentioned effects. Specifically, the highlighted text on p. 6 (lines 250–259) discusses that, as shown in Fig. 2b, the immobilization yield is more influenced by the surface topology (i.e., size, form, and porosity of the PA microparticles) than by the density of H-bond formation. For this reason, the PA12-Fe support, with its larger channel diameters, absorbs 2–3 times more pectinase compared to the neat PA12, PA6, and PA4-based systems.

Additionally, the text on p. 10 (lines 409–415), referring to the kinetic experiments in Figure 6 and Table 2, explains that for three systems (free Pec, Pec@PA4, and Pec@PA6), a combined effect of diffusion-controlled substrate supply and substrate inhibition is observed. For the remaining four systems, pure substrate inhibition likely occurs. These findings originate logically from the fitting of the experimental kinetic curves in Figure 6 and are further explained by the different morphologies of the PA supports, as detailed in Section 2.2 of the original manuscript.

In our opinion, the original text sufficiently addresses these points, and no further additions are necessary.

Q8. Materials and methods: It is a purified or a crude enzyme extract¿ Please, explain

A8. The reviewer is most likely referring to Viazym Clarif, a purified commercial enological pectinase solution. For further details, please refer to our answer to Q3 above.

Q9. Materials and methods: The authors should better explain how the immobilization parameters were determined (immobilized protein amount, specific activity). See and cite this reference: https://doi.org/10.1016/j.procbio.2019.11.026.

A9. In the Materials and Methods section, we have thoroughly detailed how the following immobilization parameters were controlled:

  • Section 3.4: Description of Pectinase Immobilization by Physical Adsorption (lines 722–731).
  • Section 3.5: Determination of the Total Protein Content (immobilization yield) using direct UV/VIS measurements (lines 732–741).
  • Section 3.6: Activity Assay for Free and Immobilized Pectinase (lines 743–756).

Additionally, Figure 2 and the texts to it explain the change of protein amount in the supernatants and the immobilization yields on each of the polyamide substrates. Figure 5 presents the absolute and specific activity (normalized by the total immobilized protein) as a function of support type and composition. Notably, the specific activity of the free enzyme is lower than that of most immobilized complexes.

We believe these descriptions sufficiently address the requirements for defining an immobilization process as outlined by Boudrant et al. (reference 38). This reference, suggested by Reviewer 3, has been included in the text (lines 731-732) and in the citation list of the revised manuscript. We do not see what additional details about immobilization parameters could be provided beyond this.

Reviewer 4 Report

Comments and Suggestions for Authors

The authors of this article have designed and performed very well the multi-criteria studies in the field of synthesis, research and application in wine processing of biocatalysts supported on polyamide polymers. However, some issues listed below require clarification and correction. Therefore, I recommend carrying out a revision of the article based on the authors' responses before its publication.

1. The article describes a wide-ranging, systematic study of commercial pectinase non-covalently bound to seven different micromolecular polyamide carriers. The aim of the work was select the best conditions for the synthesis and use of polyamide-adsorbed pectinase biocatalysts for effective clarification of pink must on a laboratory scale in a time from 80 to 180 minutes, which is acceptable for industrial wine production conditions.

2. The reviewed article is a continuation of research published by authors in 2024 in the journal Foods (reference [27]. Oliveira, S.C.; Dencheva, N.V.; Denchev, Z.Z. Immobilization of Enological Pectinase on Magnetic Sensitive Polyamide Micro-912 particles for Wine Clarification. Foods 2024, 13, 420).  It contains a description of extended research in relation to the previous article and many new results. In terms of methodology, content and structure, the article is well written and practically free of errors and shortcomings.

3. The reference [27] is cited only once in the manuscript, even though the article contains much content and even figures very similar to the material already published in the journal Foods. For example, Figure 3 published in the journal Foods is very similar to Figure 4 published in the peer-reviewed manuscript submitted to Molecules. In my opinion, this problem requires more detailed explanation by the authors and supplementation of citations.

4. The authors use the term "room temperature" three times - lines 331, 721 and 770 in the manuscript. This term can be quite broad, I think the authors should provide numerical limits of temperatures at which they conducted their research. This is especially important because these studies concern kinetics and catalysis.

5. The title of the article indicates that the authors study  biocatalytic reactions. In the chemical sense, catalysts change (primarily increase) the reaction rate by lowering the activation energy. In the article, I am unable to find an answer to the question of what chemical or biochemical reaction is meant.

6. In the keywords of the article (lines 24 and 25), the authors use concepts consisting of several words; e.g. “activated anionic ring-opening polymerization of lactams”;. In my opinion, keywords should be single words or concepts consisting of at most two words.

7. In point 2.5, the authors describe in quite detail the kinetic studies involving the biocatalysts they obtained. I did not find any data in the manuscript on the effect of temperature on the rate of the catalytic process and the activation energy. The clarification processes studied are of the nature of physical processes characterized by diffusion limitations. I would expect the authors to describe better the essence of biocatalysis involving the biocatalysts they synthesize.

Author Response

The authors of this article have designed and performed very well the multi-criteria studies in the field of synthesis, research and application in wine processing of biocatalysts supported on polyamide polymers. However, some issues listed below require clarification and correction. Therefore, I recommend carrying out a revision of the article based on the authors' responses before its publication.

We appreciate the detailed review and feedback from Reviewer 4, which have been very helpful in enhancing the quality of our manuscript. Below, we provide a comprehensive, point-by-point answer (A) to each query (Q).

Q1. The article describes a wide-ranging, systematic study of commercial pectinase non-covalently bound to seven different micromolecular polyamide carriers. The aim of the work was select the best conditions for the synthesis and use of polyamide-adsorbed pectinase biocatalysts for effective clarification of pink must on a laboratory scale in a time from 80 to 180 minutes, which is acceptable for industrial wine production conditions.

A1. This statement does not require further explanations or discussion.

Q2. The reviewed article is a continuation of research published by authors in 2024 in the journal Foods (reference [27]. Oliveira, S.C.; Dencheva, N.V.; Denchev, Z.Z. Immobilization of Enological Pectinase on Magnetic Sensitive Polyamide Micro-912 particles for Wine Clarification. Foods 2024, 13, 420).  It contains a description of extended research in relation to the previous article and many new results. In terms of methodology, content and structure, the article is well written and practically free of errors and shortcomings.

A2. This statement does not require further explanation or discussion. We sincerely thank Reviewer 4 for the recognition and appreciation of our work.

Q3. The reference [27] is cited only once in the manuscript, even though the article contains much content and even figures very similar to the material already published in the journal Foods. For example, Figure 3 published in the journal Foods is very similar to Figure 4 published in the peer-reviewed manuscript submitted to Molecules. In my opinion, this problem requires more detailed explanation by the authors and supplementation of citations.

A3. Figure 4a in the present manuscript illustrates the chemical reaction underlying the activity assay and is indeed similar to Figure 3a in [27], as the reaction mechanism is identical. However, Figure 4b in the present manuscript differs, as it excludes the UV-VIS spectra comparison, which has now been moved to the Supplementary Materials (Figure S4). Additionally, the slope, intercept, and the R2 values of the linear calibration graph in Figure 4b differ slightly from those in Figure 3b of [27], as these are distinct calibration curves obtained at different times. Given that both figures are our original work, we do not find it necessary to cite [27] again or alter the figure further to avoid similarity. We trust that the esteemed reviewer will understand this perspective.

Q4. The authors use the term "room temperature" three times - lines 331, 721 and 770 in the manuscript. This term can be quite broad, I think the authors should provide numerical limits of temperatures at which they conducted their research. This is especially important because these studies concern kinetics and catalysis.

A4. We acknowledge this criticism. In the revised manuscript, "room temperature" has been replaced with the specific value of 23 ± 1 ºC throughout the text for clarity and consistency.

Q5. The title of the article indicates that the authors study biocatalytic reactions. In the chemical sense, catalysts change (primarily increase) the reaction rate by lowering the activation energy. In the article, I am unable to find an answer to the question of what chemical or biochemical reaction is meant.

A5. Wine clarification is a biochemical process involving the removal of pectic substances, commonly referred to as pectin. Pectin is a complex polysaccharide, with polygalacturonic acid as its primary building block. In winemaking, clarification is related to the enzymatic depolymerization of polygalacturonic acid into its monomer, D-galacturonic acid, catalyzed by polygalacturonase (E.C. 3.2.1.15), a type of pectinase. This enzyme, along with pectin lyase and pectin methylesterase, is present in all enological preparations used for clarification.

D-galacturonic acid is a reducing sugar, and its concentration can be measured using the DNS method. Its chemistry is illustrated in Fig. 4b and described in the accompanying text of Section 2.4. of the present manuscript. As explained there, pectinolytic activity is quantified by UV-VIS, measuring the concentration of 3-amino-5-nitrosalicylic acid (3,5-ANS), a colored product of the 3,5-dinitrosalicylic acid (DNS) reduction, released per unit of time using the calibration curve in Fig. 4b.

Lines 298–303 of the revised manuscript summarize the essence of this information. The texts about the three pectinase enzymes already existed in lines 53-55 of the original manuscript. We trust these explanations adequately addresses the question raised.

Q6. In the keywords of the article (lines 24 and 25), the authors use concepts consisting of several words; e.g. “activated anionic ring-opening polymerization of lactams”;. In my opinion, keywords should be single words or concepts consisting of at most two words.

A6. We acknowledge this criticism and have revised the keywords accordingly, incorporating the suggested changes. Additionally, we have included one additional keyword to enhance the manuscript's accessibility.

Q7. In point 2.5, the authors describe in quite detail the kinetic studies involving the biocatalysts they obtained. I did not find any data in the manuscript on the effect of temperature on the rate of the catalytic process and the activation energy. The clarification processes studied are of the nature of physical processes characterized by diffusion limitations. I would expect the authors to describe better the essence of biocatalysis involving the biocatalysts they synthesize.

A7. As explained in our response to Q5, clarification is a biochemical, not a physical, process. The kinetic studies are detailed in Section 3.7 of the original manuscript. Specifically, pectin degradation to D-galacturonic acid was measured at various pectin concentrations (0.05–5 mg/mL) under the conditions of the pectinase activity assay described in Section 3.6, i.e., incubation for 1 min at 50ºC and then heating for 10 min in boiling water bath. These conditions align with the Miller test as described in Ref. 30.

In this study, determining the activation energy that would require performing the DNS assay at different temperatures was not within the scope. Lines 409–415 discusses the output of the kinetic experiments shown in Figure 6 and Table 2. For three systems (free Pec, Pec@PA4, and Pec@PA6), a combined effect of diffusion-controlled substrate supply and substrate inhibition is observed. For the remaining four systems, pure substrate inhibition is more likely to occur. These findings logically derive from the fitting of experimental kinetic curves in Figure 6 and are explained by the varying morphologies of the PA supports, as detailed in Section 2.2 of the manuscript.

No changes of the manuscript were made based on the above discussion.

Round 2

Reviewer 1 Report

Comments and Suggestions for Authors

The authors replied to all my comments and improved the quality of the manuscript. I suggest accepting in the present form.

Reviewer 3 Report

Comments and Suggestions for Authors

The manuscript was revised, as suggested. Therefore, I recommend its acceptacnce for publication.

Reviewer 4 Report

Comments and Suggestions for Authors

The authors have commented on all the points raised in my review and have corrected the manuscript in the appropriate places. The responses to comments show that the biocatalytic reaction rate has diffusion limitations. After reading the entire text of the corrected manuscript 2, I believe that the article can be published in Molecules.